# Tensor Network Python (TeNPy) version 1

Johannes Hauschild[1,2,⋆], Jakob Unfried[1,2,†], Sajant Anand[3], Bartholomew Andrews[3,4], Marcus Bintz[5], Umberto Borla[6], Stefan Divic[3], Markus Drescher[1,2], Jan Geiger[1,2,7], Martin Hefel[1,2], Kévin Hémery[1], Wilhelm Kadow[1,2], Jack Kemp[5], Nico Kirchner[1,2], Vincent S. Liu[5], Gunnar Möller[8], Daniel Parker[3,5,9], Michael Rader[10,11], Anton Romen[1,2], Samuel Scalet[1,12], Leon Schoonderwoerd[8], Maximilian Schulz[13,14], Tomohiro Soejima[3,5], Philipp Thoma[1,2], Yantao Wu[3,15], Philip Zechmann[1,2], Ludwig Zweng[1,2], Roger S. K. Mong (蒙紹璣)[16], Michael P. Zaletel[3,17] and Frank Pollmann[1,2]

**1** Technical University of Munich, TUM School of Natural Sciences, Physics Department, 85748 Garching, Germany
**2** Munich Center for Quantum Science and Technology (MCQST), Schellingstr. 4, 80799 München, Germany
**3** Department of Physics, University of California, Berkeley, CA 94720, USA
**4** Institute for Theoretical Physics, ETH Zürich, 8093 Zürich, Switzerland
**5** Department of Physics, Harvard University, Cambridge, Massachusetts 02138, USA
**6** Racah Institute of Physics, The Hebrew University of Jerusalem, Givat Ram, Jerusalem 91904, Israel
**7** Max-Planck-Institut für Quantenoptik, 85748 Garching, Germany
**8** Physics and Astronomy, University of Kent, Ingram Building, Canterbury CT2 7NZ, United Kingdom
**9** Department of Physics, University of California, San Diego, CA 92093, USA
**10** Institute for Theoretical Physics, University of Innsbruck, 6020 Innsbruck, Austria
**11** Fraunhofer Austria Research GmbH, 6112 Wattens, Austria
**12** Department of Applied Mathematics and Theoretical Physics, University of Cambridge, Cambridge CB3 0WA, United Kingdom
**13** SUPA, School of Physics and Astronomy, University of St Andrews, North Haugh, St Andrews, Fife KY16 9SS, United Kingdom
**14** Max Planck Institute for the Physics of Complex Systems, Nöthnitzer Str. 38, 01187 Dresden, Germany
**15** RIKEN iTHEMS, Wako, Saitama 351-0198, Japan
**16** Department of Physics and Astronomy, University of Pittsburgh, Pittsburgh, PA 15260, USA
**17** Material Science Division, Lawrence Berkeley National Laboratory, Berkeley, CA 94720, USA

⋆ johannes.hauschild@tum.de , † jakob.unfried@tum.de

## Abstract

**TeNPy (short for 'Tensor Network Python') is a** `python` **library for the simulation of strongly correlated quantum systems with tensor networks. The philosophy of this library is to achieve a balance of readability and usability for new-comers, while at the same time providing powerful algorithms for experts. The focus is on MPS algorithms for 1D and 2D lattices, such as DMRG ground state search, as well as dynamics using TEBD, TDVP, or MPO evolution. This article is a companion to the recent version 1.0 release of TeNPy and gives a brief overview of the package.**

## 1 Introduction

Large scale numerical simulations play an essential role in understanding strongly correlated quantum matter. Tensor network methods, and in particular methods based on matrix product states (MPS) [1], have become a state-of-the-art tool to investigate these systems, especially in fermionic or frustrated settings, where the infamous sign problem inhibits most Monte Carlo simulation techniques.

For local, gapped Hamiltonians in 1D, the area law of entanglement [2] guarantees an efficient representation of the ground state as an MPS and has enabled the success of the density matrix renormalization group (DMRG) method [3,4] to find these ground states numerically. It has been demonstrated that DMRG ground state searches are an effective tool also in a broader setting of long-range-interacting [5] or critical [6,7] systems, and can additionally be applied to study 2D models, via simulation on cylinder geometries [8]. Dynamics can be efficiently simulated with the time-evolving block decimation (TEBD) [9], matrix product operator (MPO) evolution [10], and time-dependent variational principle (TDVP) [11,12] methods, as long as the entanglement of the evolved state remains manageable. Various other areas of application include the simulation of thermal states, non-equilibrium dynamics, excitation spectra and many more. The performance and accuracy of these methods can be significantly improved by exploiting conservation laws induced by Abelian [13,14] or non-Abelian [15,16] symmetries. In this article, we do not provide detailed introductions for this plethora of methods, concepts and algorithms and instead refer the interested reader to the existing reviews [17–20].

The TeNPy package for tensor network simulations is based on the codes used in references [21,22], and was first introduced as an open-source `python` package in reference [23]. It aims to provide a flexible, easy-to-use and well-documented interface for tensor network algorithms suitable for beginners, while enabling high-performance simulations for experts. The modular package structure allows users to visit the nested layers of complexity as needed

– ranging from a high-level viewpoint, e.g. establishing a phase diagram, concrete MPS algorithms and their various parameters, through to the underlying linear algebra of individual tensors. A particular focus is placed on user-friendly initialization of tensor networks that does not require technical knowledge, or indeed any interaction with individual tensor entries. This includes generating MPS from product states, singlet coverings or random unitary evolution, as well as the MPO representation of a particular Hamiltonian [24, 25] or its (approximate) time evolution operator [10]. This allows users to specify models in the natural language of couplings between sites on a given lattice geometry and, in a few lines of either python code or yaml configuration files, perform e.g. an efficient DMRG simulation which takes care of (i) the mapping between the 1D linear geometry of the MPS and the physical lattice, (ii) the canonical form of the MPS and finding the local ground state using an iterative eigensolver, and (iii) exploiting Abelian symmetries of the model using block-sparse linear algebra.

TeNPy is open-source and publicly maintained on GitHub[1] and has extensive online documentation[2], as well as an active community forum[3].

This article accompanies the recent version 1.0 release of TeNPy and is structured as follows. First, we give a broad overview of the package and its features in section 2. In sections 3 and 4, we showcase concrete example simulations in code snippets, before benchmarking the performance in section 5. Finally, we give details about the format in which TeNPy tensors are stored to file in section 6, and conclude with an outlook over current and future development directions in section 7.

## 2  Package overview

The TeNPy package is comprised of submodules which can be conceptualized as layers of abstraction. In this section, we introduce the main submodules and their most relevant contents from the high-level user-friendly simulation module to the low-level linalg module. The structure is visually summarized in figure 1.

The tenpy.simulations module offers a high-level interface for entire simulations, from a few input parameters, such as coefficients in the Hamiltonian, system size, bond dimensions, etc. to a few quantities of interest, such as expectation values, entanglement entropies, and more. A simulation run consists roughly of initializing a physical model, which defines the Hamiltonian and a lattice geometry, initializing a tensor network (e.g. an MPS), running an algorithm (e.g. DMRG), performing some measurements, optional post-processing, and finally saving results to disk. As demonstrated in section 4.2, the input parameters can be specified in an easy-to-read yaml format, which also facilitates reproducibility of simulation results and longevity of input parameters. The simulations are highly flexible and, for example, allow parameters to change sequentially, such as increasing the bond dimension gradually or sweeping a parameter through the phase diagram. They can also be resumed after periodic checkpoints, which is often needed when running long simulations on computing clusters.

The tenpy.algorithms module implements algorithms to operate on tensor network states. This includes MPS compression, DMRG, TEBD, TDVP and MPO time evolution, as well as an experimental implementation of variational uniform matrix product states (VUMPS) [26]. Each algorithm, and each of its notable variations (such as two-site versus single-site DMRG) is implemented in a dedicated class, such as e.g. the SingleSiteDMRGEngine. The module contains a multi-layered hierarchy of class inheritance, such that algorithmic steps that are common between several algorithms are implemented and maintained centrally. For example,

---

[1]The TeNPy GitHub repository: https://github.com/tenpy/tenpy
[2]The TeNPy online documentation: https://tenpy.readthedocs.io/en/latest
[3]The TeNPy user forum: https://tenpy.johannes-hauschild.de

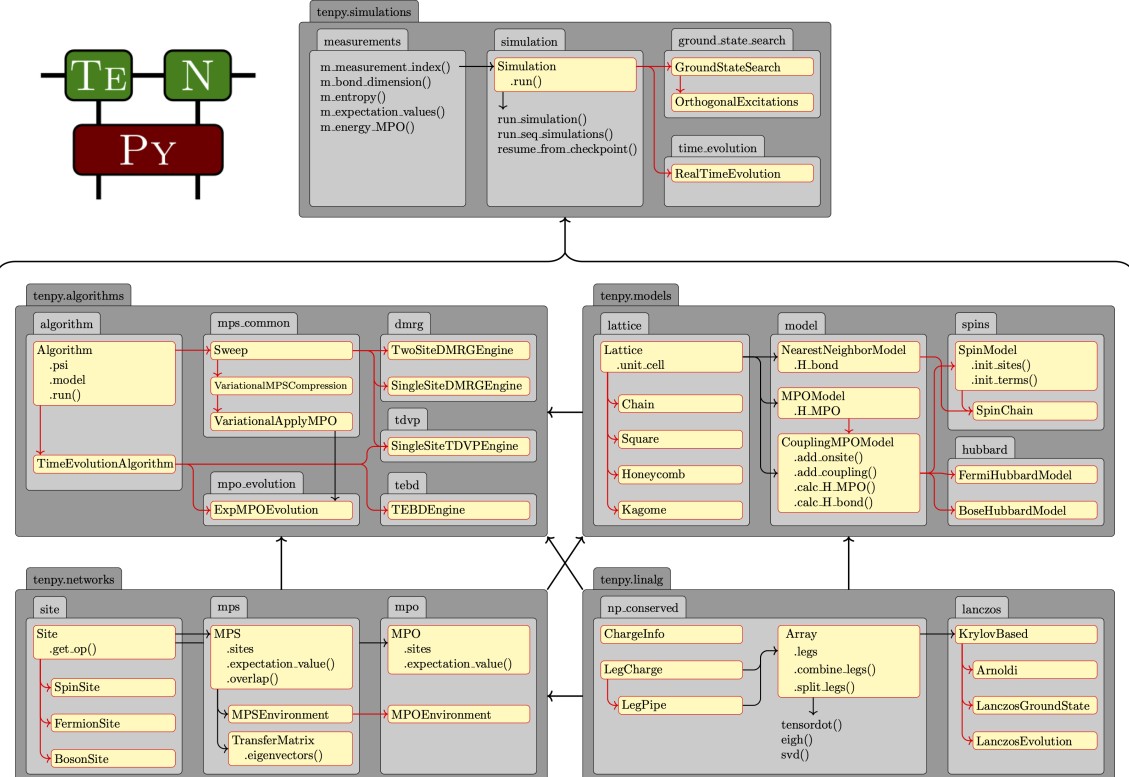

Figure 1: Overview of the most important modules, classes and functions of TeNPy. Gray backgrounds indicate modules, yellow backgrounds indicate classes. Red arrows indicate class inheritance relations, black arrows indicate direct use. There is a hierarchy from high-level (`tenpy.simulations`) to low-level (`tenpy.linalg`) functionality in the natural reading direction.

both DMRG and TDVP require computing environment tensors, which is implemented in the `Sweep` class.

The `tenpy.models` module offers two types of classes. First, the `lattice` classes that represent lattices in any number of dimensions, though one-, quasi-one- and two-dimensional systems are most common. These classes facilitate the mapping between a lattice geometry and the geometry of the tensor network, such as how to wind an MPS around a 2D cylinder. Second, the `model` classes, which implement the details of a physical model on the lattice, mostly in terms of a Hamiltonian, which can be conveniently specified in terms of couplings, e.g. between pairs of nearest neighbors of the lattice. Defining new models is thus accessible to non-experts, and does not require technical knowledge of representing Hamiltonians as tensor networks.

The `tenpy.networks` module implements classes that represent sites of a physical lattice, defining the local Hilbert spaces along with their conserved charges and local operators. Additionally, it defines classes for the supported tensor networks, such as `MPS` and `MPO`, along with a multitude of convenience functions to extract quantities of interest, such as expectation values, entanglement spectra, and many more. Creating MPO representations of Hamiltonians or their exponentials is fully automated, as is initializing a multitude of exact MPS, such as product states, singlet coverings, etc.

The `tenpy.linalg` module implements numerical linear algebra that (optionally) exploits charge conservation induced by arbitrary Abelian symmetry groups. It offers the `Array` class, representing symmetric tensors with a block-sparse structure along with linear alge-

bra operations on them, such as tensor contraction, eigendecomposition, SVD, etc. As such, `tenpy.linalg.np_conserved` fulfills a similar role as the popular `numpy` package, albeit with a smaller scope of functions offered. Additionally, iterative eigensolvers, such as the Lanczos algorithm commonly used in DMRG, are implemented for linear operators which act on the block-sparse `Arrays`. The popular NCON-style [27] for specifying contractions of multiple tensors is implemented as well, as `tenpy.ncon`.

## 3 Example: Dynamics in a Heisenberg chain

In this section, we present an example use-case, simulating non-equilibrium dynamics in a spin-$\frac{1}{2}$ Heisenberg chain

$$H = J \sum_{\langle i,j \rangle} \vec{S}_i \cdot \vec{S}_j, \tag{1}$$

where $\vec{S}$ is the vector of spin operators $S^\alpha = \sigma^\alpha/2$. We walk through a TEBD simulation – approximating the time evolution $|\psi(t)\rangle = e^{-iHt} |\psi(t=0)\rangle$ and extracting time-dependent observables – in `python` code snippets. The code is available in the GitHub repository[4].

TeNPy can be installed[5] from the `pip` and `conda` package managers. When installed, we can import the package as

```
1 import tenpy
```

First, we configure the model (1) on a 1D chain with $L = 50$ sites, setting $J = 1$.

```
2 model_params = dict(L=50, Jx=1, Jy=1, Jz=1)
3 model = tenpy.SpinChain(model_params)
```

The boundary conditions are open by default, and make TeNPy use finite MPS. By default, models enforce conservation of the largest (Abelian) symmetry for the given model parameters, in this case the $U(1)$ symmetry that conserves $Q = \sum_i S_i^z$. This can be disabled by adding `conserve=None` to the model parameters, or relaxed to a smaller symmetry by adding, for example, `conserve='parity'` to only enforce the $\mathbb{Z}_2$ subgroup conserving the parity $\tilde{Q} = \sum_i (S_i^z + 1/2) \mod 2$ of the number of up-spins. There is a full index of these options and their possible values in the online documentation[6].

Second, we initialize the Néel state $|\psi(t=0)\rangle = |\uparrow\downarrow\uparrow\downarrow ...\rangle$,

```
4 psi = tenpy.MPS.from_lat_product_state(
5   model.lat, [['up'], ['down']]
6 )
```

and a TEBD engine,

```
7  engine_params = dict(
8    dt=0.01,
9    N_steps=5,
10   trunc_params=dict(
11     chi_max=50,
12     svd_min=1e-10,
13   ),
14 )
15 engine = tenpy.TEBDEngine(psi, model, engine_params)
```

---

[4]https://github.com/tenpy/tenpy/tree/main/examples/v1_publication.

[5]To install TeNPy with a package manager, run `conda install conda-forge::physics-tenpy` or `pip install physics-tenpy` or see detailed installation instructions in the documentation https://tenpy.readthedocs.io/en/latest/INSTALL.html.

[6]https://tenpy.readthedocs.io/en/latest/reference/tenpy.models.spins.SpinModel.html#cfg-config-SpinModel for the options of the `SpinChain`, or see the full index at https://tenpy.readthedocs.io/en/latest/cfg-option.html.

which performs five steps of Trotterized time evolution, each of size $dt = 0.01$, per call of its `.run()` method. In this example, it is configured with a maximum bond dimension of $\chi = 50$ and discards singular values below $10^{-10}$.

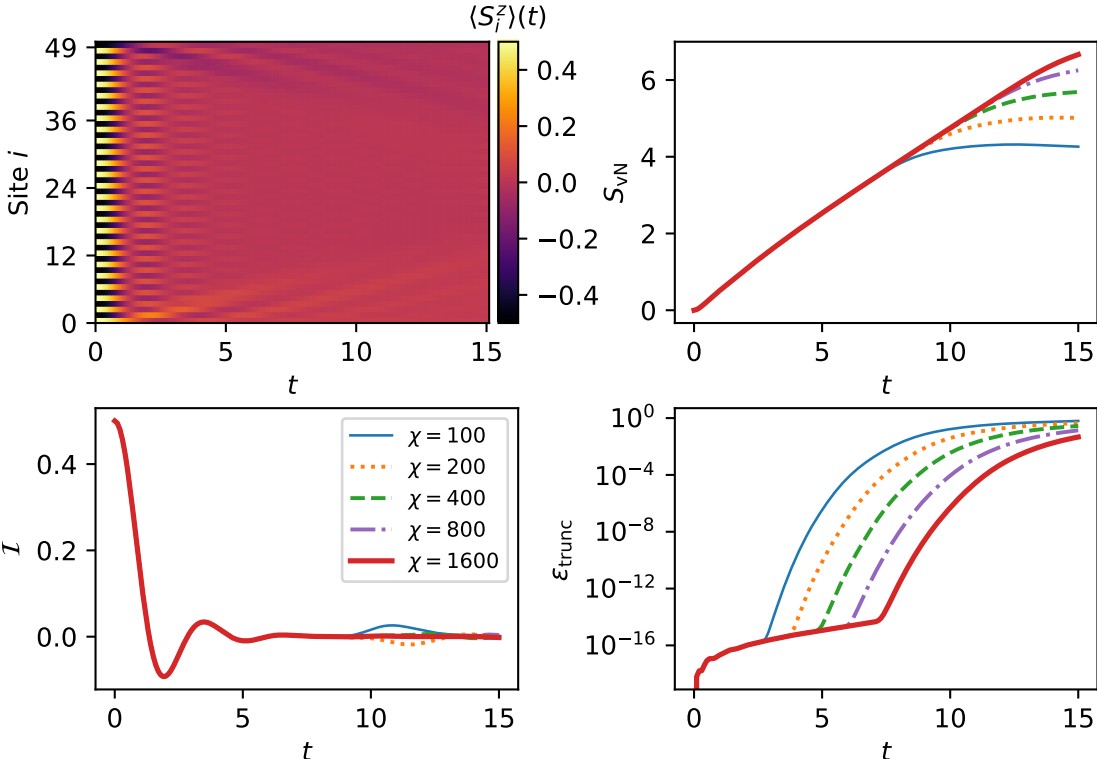

Figure 2: Dynamics of the Heisenberg chain (1) starting from a Néel state at $t = 0$. The top left panel shows the local $z$-magnetization (at $\chi = 1600$), which equilibrates to a uniform value over time, as quantified by the sublattice imbalance (2) in the bottom left panel. The right panels show the half-chain entanglement entropy $S_{\mathrm{vN}}$ and cumulative truncation error, respectively. Note that the plots show data for time scales that have a large truncation error, even at the largest bond dimension, and should be disregarded for analysis. We include them here to showcase the saturating behavior of the entanglement entropy.

To run the simulation, we perform the following steps within an outer loop over multiple time steps: Let the engine evolve the state `psi`, modifying it in-place, and then measure observables.

```
t = [0]
S = [psi.entanglement_entropy()]
Sz = psi.expectation_value('Sz')
mag_z = [Sz]
imbalance = [sum(Sz[::2] - Sz[1::2]) / psi.L]
for n in range(100):
  engine.run()  # evolves psi in-place
  t.append(engine.evolved_time)
  S.append(psi.entanglement_entropy())
  Sz = psi.expectation_value('Sz')
  mag_z.append(Sz)
  imbalance.append(sum(Sz[::2] - Sz[1::2]) / psi.L)
```

Here, `imbalance` accumulates the sublattice imbalance

$$\mathcal{I} = \frac{1}{L} \sum_j (-1)^j \langle S_j^z \rangle = \frac{1}{L} \left( \sum_{j \in A} \langle S_j^z \rangle - \sum_{j \in B} \langle S_j^z \rangle \right) \tag{2}$$

at every time step.

The results are shown in figure 2. As expected for a global quench [28], entanglement grows linearly with time, such that accurate simulation at bond dimension $\chi$ is possible only to a time scale $\sim \log \chi$ and doubling the bond dimension only pushes that time scale by a constant.

## 4 Example: DMRG study of the 2D Ising model

In this section, as a second example, we characterize the ground state phase diagram of the transverse field Ising model on an infinite cylinder square lattice, using DMRG. A more in-depth guide to DMRG simulations in TeNPy is available in the documentation[7].

### 4.1 Python code

We first walk through performing the simulation in `python` code snippets. We configure the spin-$\frac{1}{2}$ transverse field Ising model on a 2D square lattice. The following lines initialize the model with the Hamiltonian

$$H = -J \sum_{\langle i,j \rangle} \sigma_i^x \sigma_j^x - g \sum_i \sigma_i^z, \tag{3}$$

where $\sigma^\alpha$ are Pauli matrices with eigenvalues $\pm 1$. The boundary conditions correspond to an infinite cylinder. In the $y$-direction, the system is periodic with circumference $L_y$. In the $x$-direction the simulated system is infinite and we choose a unit cell of width $L_x = 2$ (with $L_x L_y = 2L_y$ sites in total) for the MPS simulation.

```python
import tenpy
g = 1.42  # example transverse field strength
Ly = 4  # example cylinder circumference
model_params = dict(
  bc_MPS='infinite', bc_y='cylinder',
  lattice='Square', Lx=2, Ly=Ly,
  J=1, g=g,
  conserve='best',  # conserve parity
  # conserve='None',  # conserve nothing
)
model = tenpy.TFIModel(model_params)
```

The largest Abelian symmetry for this model is the $\mathbb{Z}_2$ symmetry which conserves the parity $Q = \sum_i (\sigma_i^z + 1)/2 \mod 2$ of the number of up-spins, and is conserved by default.

Second, we need to provide an initial state from which to start the simulation. We choose a Néel product state.

```python
psi = tenpy.MPS.from_lat_product_state(
  model.lat, [[['up'], ['down']]]
)
```

Note that when using charge conservation, the initial guess determines the charge sector of the target state.

We now configure a DMRG engine with density matrix perturbations [29] ("mixer"), maximum bond dimension of 200 and discarding singular values below $10^{-10}$.

---

[7]https://tenpy.readthedocs.io/en/latest/intro/dmrg-protocol.html

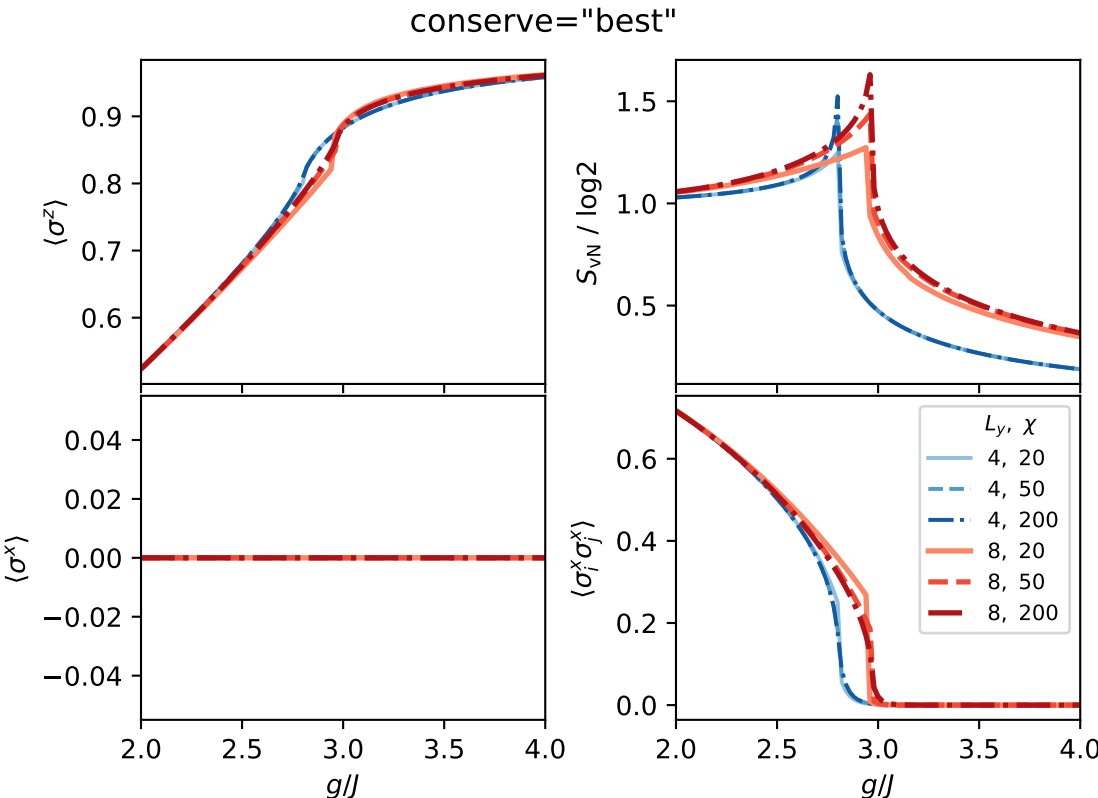

Figure 3: Phase diagram of the transverse field Ising model (3), obtained using DMRG on an infinite cylinder with circumference $L_y$ (color) and bond dimension $\chi$ (line style) with parity conservation enforced. The left panels show expectation values of $\sigma^z$ and $\sigma^x$, averaged over the unit cell. The top right panel shows the von Neumann entanglement entropy for a cut along a unit cell boundary. The bottom right panel shows the correlation function $\langle \sigma_i^x \sigma_j^x \rangle$, where site $j = i + 10\vec{a}_x$ is ten unit cells to the right (along the cylinder axis) of site $i$.

```
15  dmrg_params = dict(
16    mixer=True,
17    trunc_params=dict(
18      chi_max=200,
19      svd_min=1e-10,
20    ),
21  )
22  engine = tenpy.TwoSiteDMRGEngine(
23    psi, model, dmrg_params
24  )
```

We can now run DMRG to obtain the ground state approximation `psi` and its `energy`. The `engine.run()` method executes the numerically demanding simulation and emits status messages to the logging system, configurable via `tenpy.setup_logging`. Finally, we evaluate some observables in the converged ground state.

```
25  energy, psi = engine.run()
26  mag_z = numpy.average(psi.expectation_value('Sz'))
27  mag_x = numpy.average(psi.expectation_value('Sx'))
28  corr_xx = psi.correlation_function(
29      'Sx', 'Sx', [0], [10 * model.lat.N_sites]
30  )
31  entropy = psi.entanglement_entropy()[0]
```

An extended version of this example is available in the GitHub repository[8]. It includes an outer loop over a grid of values $g$ for the transverse field, and generates the phase diagrams in figures 3 and 4.

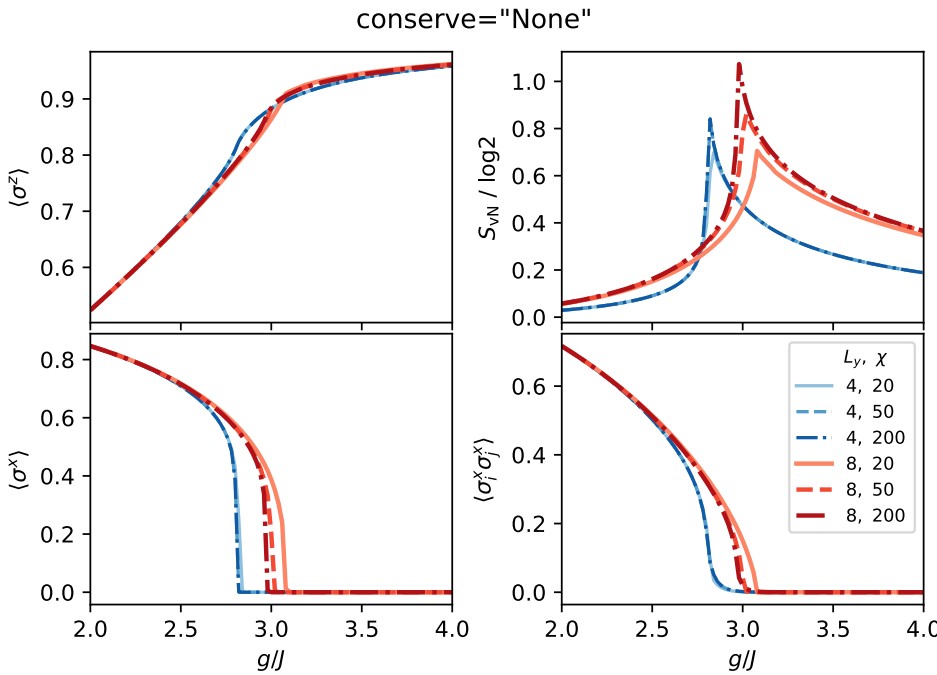

Figure 4: Phase diagram of the transverse field Ising model (3), analogous to figure 3, except simulated without symmetry conservation.

As expected, we observe a phase transition from a $\sigma^x$ ordered phase for $g < g_c \approx 3$ to a disordered phase at large fields. Increasing the cylinder circumference shifts the position $g_c$ of the critical point and increases the entanglement entropy proportionally to $L_y$, since

---

[8]https://github.com/tenpy/tenpy/tree/main/examples/v1_publication

the ground state is a 2D area law state. This in turn means that higher bond dimensions, *exponential* in $L_y$, are required to simulate larger cylinders to the same accuracy. This can be seen in the data, as $\chi = 50$ is already sufficient for the $L_y = 4$ cylinder and further increasing $\chi$ does not improve results, while $\chi = 200$ is still not converged for the $L_y = 8$ cylinder.

If the $\mathbb{Z}_2$ symmetry is enforced, as in figure 3, DMRG in the ordered phase converges to a cat state, that is an equal-weight superposition of the two "maximally ordered" states with maximal and minimal order parameter, such that $\langle \sigma^x \rangle$ averages to zero. For infinite MPS, this can be detected by a degenerate dominant eigenvector of the transfer matrix. As a result, the order does not manifest in $\langle \sigma^x \rangle$, only in the correlation function $\langle \sigma_i^x \sigma_j^x \rangle$, and the entanglement entropy is increased by **log 2** in the ordered phase. Simulating without enforcing the symmetry converges to a maximally ordered state, such that these characteristics are not present in figure 4.

## 4.2 Simulation with YAML config

Instead of explicitly looping in `python` code, we can configure such a phase diagram sweep using the `yaml` config input format.

```yaml
#!/usr/bin/env -S python -m tenpy

simulation_class : GroundStateSearch

model_class: TFIModel
model_params:
  bc_MPS: infinite
  bc_y: cylinder
  lattice: Square
  Lx: 2
  Ly: 4
  J: 1
  g: !py_eval |
    np.linspace(2, 4, 101, endpoint=True)

initial_state_params:
  method: lat_product_state
  product_state: [[[up], [down]]]

algorithm_class: TwoSiteDMRGEngine
algorithm_params:
  mixer: True
  trunc_params:
    svd_min: 1.e-10
    chi_max: 258

connect_measurements:
  - - tenpy.simulations.measurement
    - m_onsite_expectation_value
    - opname: Sz

sequential:
  recursive_keys:
    - model_params.g

directory: results
output_filename_params:
  prefix: dmrg_tfi_cylinder
  parts:
    model_params.g: 'g_{0:.1f}'
  suffix: .h5
```

Lines 3–30 configure a simulation that is equivalent to the `python` snippets of the previous section, except that we have already passed a grid of multiple values to the g model parameter. The remaining lines then indicate that the simulation should repeat for each value of that grid, and specifies the filename for outputs as a function of the current g value. With the header in line 1, the `yaml` config is executable and will run the simulation. Alternatively, the `tenpy-run` command line interface can be used to run such simulations. For a full specification of the `yaml` input format, refer to the documentation[9].

## 5 Benchmarks

There are often raised concerns that `python` is slow compared to compiled languages like C++ or just in time compiled languages like `Julia`. However, we find that this is not an issue for simulations based on tensor networks: when the sizes of the individual tensors are scaled up by control parameters like the bond dimension $\chi$ in an MPS, the computation time quickly becomes dominated by individual tensor operations, namely tensor contractions, singular value and eigendecompositions, and reshaping of the tensors by transposing, combining or splitting legs. For theses operations, ultimately the highly optimized BLAS and LAPACK libraries (or variants thereof, like Intel's MKL) are called. On this level, the tensor network algorithms like DMRG are high-level algorithms that merely handle the order of the cost-dominating tensor operations and check convergence criteria.

While the parameters for convergence criteria have an obvious impact on the overall runtime by controlling the length of outer loops (like the number of sweeps in DMRG or the number of iterations in the Lanczos eigensolver), we fix them in figure 5 for a fair comparison between TeNPy and the ITensor library [30, 31]. We find comparable speed[10], with the outer parameters and which underlying BLAS library one links against often having a bigger influence on the overall runtime than the choice between TeNPy and ITensor. In particular, we find the same asymptotic scaling $\mathcal{O}(\chi^3)$ and find that the overhead of using `python` is negligible when the total runtime per sweep exceeds minutes.

In the current version 1, TeNPy only supports the conservation of the Abelian $U(1)$ subgroup of the $SU(2)$ symmetry, enabled by the model parameter `conserve='Sz'`. In that case, the tensors are internally split up into smaller blocks, with charge conservation dictating which blocks need to be contracted. This leads to some overhead (especially in reshaping operations) that is visible at small bond dimension, but significantly speeds up the calculation at larger bond dimensions $\chi \gtrsim 100$. This overhead is also visible as a plateau of the charge-conserving data in the bottom panel of figure 5, when the maximum number of blocks in each tensor is reached, but the individual block sizes are small.

Since TeNPy itself does not utilize multi-threading, we can only find speed-ups by using multiple cores when the individual blocks handed to the BLAS and LAPACK library are of significant size. This is clearly visible in the bottom panel of figure 5 when the curves for different number of cores only deviate when block sizes become significant. For the DMRG calculations with charge conservation and $\chi \lesssim 1000$, we find that increasing the number of cores beyond 4 comes with significantly diminished returns. A good indicator is to check whether the asymptotic scaling $\propto \chi^3$ was reached – if this is not the case, using more cores will not significantly speed up the simulation.

---

[9]https://tenpy.readthedocs.io/en/latest/intro/simulations.html

[10]Our latest benchmark results and the code to reproduce them are available at https://github.com/tenpy/tenpy_benchmarks. A similar comparison is available at https://itensor.github.io/ITensorBenchmarks.jl/dev/tenpy_itensor/index.html.

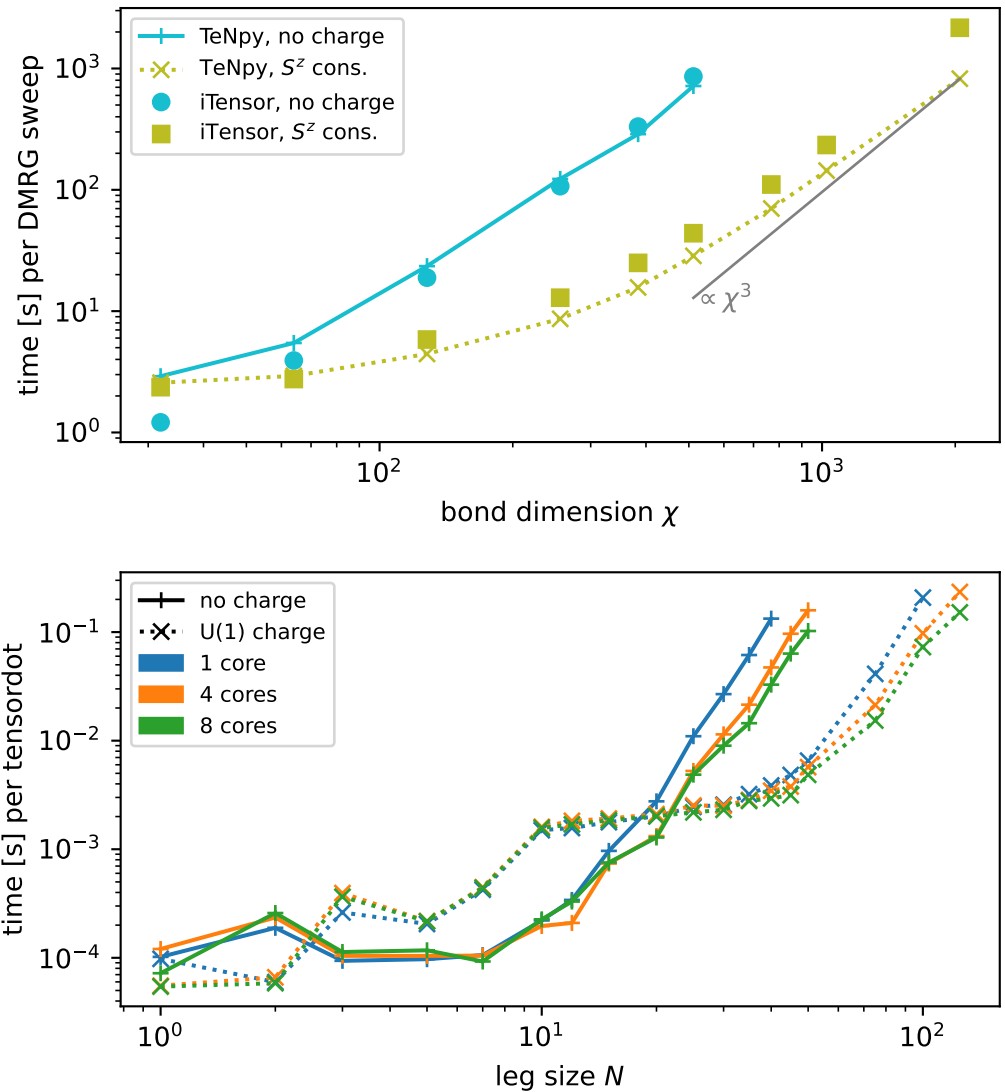

Figure 5: Benchmarks showing the walltime in seconds against a control parameter of the problem size. The top panel shows the time per DMRG sweep for an $N = 100$ sites $S = 1$ Heisenberg chain with a fixed number of three iterations in the Lanczos eigensolver. We find comparable speed and in particular the same scaling as the ITensor library (version 0.6.16) [30,31]. The bottom panel shows the time for a single `tensordot` call, contracting two legs of random tensors of shape $(N, N, N, N)$. For the case of the $U(1)$ charge, each leg is divided into 10 charge sectors, splitting the tensor into smaller blocks. This yields a plateau starting at $N = 10$, with a regime of a maximal number of blocks per tensor but tiny individual block sizes. Benchmarks were performed on an Intel Xeon 8368 CPU, linked against Intel's MKL library, using a single core in the top panel, and color indicating the number of cores utilized in the bottom panel.

# 6 Storage Specification for Symmetric Tensors

In this section, we specify the format in which TeNPy saves data to disk in the HDF5 file format[11]. It implements a custom toolbox in the `tenpy.tools.hdf5_io` module to store python objects to file, if they are either supported builtin types, a numpy array or implement the interface defined by `tenpy.Hdf5Exportable`. For example, entire MPS states of type `tenpy.MPS` may be stored in an HDF5 file, and this section is intended as a starting point to reading them out in custom software and to future-proof these datasets, making them readable even without a TeNPy installation.

HDF5 files can be thought of as file trees, i.e. nested structures of named HDF5 *groups* that have named *attributes* associated with them, and may contain named *datasets* or further groups. The `tenpy.tools.hdf5_io` module provides functions to load and save nested python data into the HDF5 format, mapping the hierarchical structure of the data into the tree structure of HDF5, while handling special cases like circular references: it deduplicates multiple references to the same object as links in HDF5. Python classes like `tenpy.MPS` or `tenpy.Array` are mapped to HDF5 groups, with the subgroups representing the class attributes. The hdf5 groups have a few attributes `"type"`, `"class"` and `"module"` that facilitate recreating the equivalent python objects.

In the following, we list the HDF5 structure of `tenpy.Array`, the type for symmetric tensors and the types that appear as subgroups for it. Let us first summarize the relevant structure of a symmetric tensor, reviewed in more detail in references [13,14,23]. We assume there is a number $N_q$ of independent Abelian symmetries that are conserved, each either a $U(1)$ or $\mathbb{Z}_N$ group. Every physical and virtual Hilbert space is organized by the resulting quantum numbers, that is each index $a$ – labelling a basis vector for a Hilbert space $\mathcal{H}$ – is associated with a concrete value $q_a \in \mathbb{Z}^{N_q}$ of the conserved charges, described by a tuple of $N_q$ integers, one for each of the symmetries. For a $U(1)$ group, the conserved charge is unrestricted, while it is in $\{0, \ldots, N-1\}$ for $\mathbb{Z}_N$. The charges are additive when forming combined spaces $\mathcal{H}^{[n]} \otimes \mathcal{H}^{[m]}$, such that a combined index $(a_n, a_m)$ has charge $q_{a_n}^{[n]} + q_{a_m}^{[m]}$, where addition is modulo $N$ for the charges of a $\mathbb{Z}_N$ symmetry. The resulting charge rule

$$\sum_{n=1}^{\ell} \zeta^{[n]} q_{a_n}^{[n]} \neq Q \implies M_{a_1, a_2, \ldots, a_\ell} = 0 \tag{4}$$

gives a necessary condition for the entries of a tensor $M$ with $\ell$ legs to be non-zero. It includes a choice of sign $\zeta^{[n]} = \pm 1$ for each leg, that facilitates treating bra-like legs that would otherwise need to have opposite charges, as well as a total charge $Q$ that allows representation of not only symmetric, i.e. charge-*conserving* tensors ($Q = 0$), but also tensors that raise/lower the conserved charge, such as e.g. boson creation operators when particle number is conserved.

---

[11]Like almost all `python` objects, TeNPy arrays can alternatively be saved in the binary pickle format, but HDF5 is preferred for a better cross-platform and cross-version compatibility.

A symmetric tensor, of type `tenpy.Array` with $\ell$ legs and $b$ non-zero blocks, is stored in the following format.

```
Array:   group
```

— `block_inds:`  2D integer dataset, shape $b \times \ell$
  Describes which elements $M_{a_1,\ldots,a_\ell}$ of the tensor are stored in which of the `blocks`.
  The entries in `blocks[i]` are those with indices $a_1,\ldots,a_\ell$ that fulfill
  `legs/j/slices[b_ij]` $\leq a_j <$ `legs/j/slices[b_ij + 1]` for all $j$, where
  `b_ij = block_inds[i, j]`.

— `blocks:`  group

  — `0:`  $\ell$ D dataset
  The first block. Type is either float or complex and should match the
  `Array/dtype`. In general for the $i$-th block `Array/blocks/i`, the shape is
  given by $(S_{0,i+1} - S_{0,i}) \times \cdots \times (S_{\ell-1,i+1} - S_{\ell-1,i})$, where $S_{n,i}$ denotes the entry
  `Array/legs/n/slices[i]`.

  — `...`
  Further blocks with integer names $1$ to $(b-1)$.

— `chinfo:`  group with `"ChargeInfo"` format
  Charge Information, specifying the symmetry(-ies). The format is specified below.

— `dtype`
  Encodes the numpy dtype of the blocks.

— `labels:`  group
  One label per leg

  — `0:`  scalar string dataset
  The label for the first leg.

  — `...`
  Further labels with integer names 1 to $(\ell-1)$.

— `legs:`  group

  — `0:`  group with `"LegCharge"` format
  Specifies the first leg. The format is specified below.

  — `...`
  Further legs with integer names $1$ to $(\ell-1)$.

— `total_charge:`  1D integer dataset, shape $N_q$
  The RHS $Q$ of the charge rule (4).

The `Array` type above stores as one of its subgroups the `ChargeInfo` type, in the structure outlined below.

```
ChargeInfo:  group
│
├── num_charges:  integer attribute
│   The number $N_q$ of conserved charges.
│
├── names:  group
│   │
│   ├── 0:  scalar string dataset
│   │   A descriptive name for the first charge.
│   │
│   └── ...
│       Further descriptive names with integer names **1** to $N_q - 1$.
│
└── U1_ZN: 1D integer dataset, shape $N_q$
    For each charge, specifies the symmetry group. **1** means $U(1)$, other $N > 1$ mean $\mathbb{Z}_N$.
```

An `Array` also contains objects of type `LegCharge`, which are stored in the following format.

```
LegCharge:  group
│
├── format:  string attribute
│   The storage format. This specification is only valid for format="blocks".
│
├── ind_len:  integer attribute
│   The dimension $\dim \mathcal{H}$ of the leg.
│
├── qconj:  integer attribute
│   Either +1 or −1. The additional $\zeta$ sign in the charge rule (4).
│
├── charges:  2D integer dataset, shape $N_c \times N_q$
│   The unique charge values that appear as $q_a$.
│   See slices for mapping indices to charges.
│
├── chinfo:  group with "ChargeInfo" format
│
└── slices:  1D integer dataset, shape $(N_c + 1)$
    Increasing integers [0, ..., ind_len], specifying the charges.
    Indices $a$ with slices[i] <= $a$ < slices[i + 1] have charge $q_a$ = charges[i].
```

## 7  Future Directions

An extension of the `tenpy.linalg` module that handles linear algebra of symmetry-conserving tensors is currently under active development, publicly on GitHub[12]. The goal is to generalize both the symmetry structure of the tensors, as well as the storage type of the dense blocks. As a result, the new `linalg` module can enforce, in addition to Abelian symmetries, also

---

[12]Current development is in the `v2_alpha` branch of the GitHub repository at https://github.com/tenpy/tenpy.

non-Abelian symmetries, simulate fermionic or anyonic degrees of freedom, or combinations thereof. Additionally, the storage type of the dense blocks holding the free parameters of the tensors can be adjusted in a modular fashion. This enables not only hardware acceleration on GPU, but also advanced functionality of packages like PyTorch, which can then directly perform automatic differentiation on TeNPy functions, or use them as trainable layers in neural networks. At the time of writing, a full prototype for this new functionality exists and remains to be optimized for performance.

Additionally, support for mixed state and Heisenberg evolution of operators is currently in development. This will include support for Lindbladian evolution to extend TeNPy to open quantum systems.

Future directions for the library may include broader classes of tensor networks, such as the multi-scale entanglement renormalization ansatz (MERA), projected entangled pair states (PEPS) and isometric tensor network states (isoTNS), as well as providing a user-friendly interface for the simulation of quantum circuits using the existing TEBD implementation.

## Acknowledgements

We are grateful for many fruitful discussions, in particular with Mari Carmen Bañuls, Jens Bardarson, Ignacio Cirac, Philippe Corboz, Jan von Delft, Matthew Fishman, Bernhard Jobst, Claudius Hubig, Michael Knap, Ying-Jer Kao, Jonas Kjäll, Andreas Läuchli, Jutho Haegeman, Ian McCulloch, Sebastian Paeckel, Ananda Roy, Tibor Rakovszky, Ulrich Schollwöck, Philipp Schmoll, Norbert Schuch, Miles Stoudenmire, Luca Tagliacozzo, Laurens Vanderstraeten, Ruben Verresen, Frank Verstraete, and Guifré Vidal.

**Author contributions** The code is based on previous (non-public) codes written by F.P., M.P.Z. and R.M., together with J.H. Bardarson and J.A. Kjäll [21, 22]. J.H. used this code as a basis for a rewrite into the structure of the current open-source library [23]. J.H. and J.U. are currently the main developers; code contributions of the remaining authors are available in the git history of the code repository.

**Funding information** J.U., N.K. and F.P. acknowledge support by the European Research Council (ERC) under the European Union's Horizon 2020 research and innovation program under Grant Agreement Nos. 771537, 851161, and 101158022. F.P. acknowledges the support of the Deutsche Forschungsgemeinschaft (DFG, German Research Foundation) under Germany's Excellence Strategy EXC-2111-390814868. F.P.'s and J.H.'s research is part of the Munich Quantum Valley, which is supported by the Bavarian state government with funds from the Hightech Agenda Bayern Plus. B.A. acknowledges support from the Swiss National Science Foundation under grant nos. P500PT_203168 and P5R5PT_225346. M.P.Z., S.A. and B.A. were supported by the U.S. Department of Energy, Office of Science, Office of Basic Energy Sciences, Materials Sciences and Engineering Division under contract no. DE-AC02-05-CH11231 (Theory of Materials program KC2301). G.M. acknowledges funding from the Royal Society via University Research Fellowship awards UF120157 and URF\R\180004, and from the Engineering and Physical Sciences Research Council under grant no. EP/P022995/1. L.S. acknowledges funding from a Vice Chancellor's Scholarship of the University of Kent. Y.W. acknowledges funding from the RIKEN iTHEMS fellowship and Japan Science and Technology Agency (JST) as part of Adopting Sustainable Partnerships for Innovative Research Ecosystem (ASPIRE), Grant Number JPMJAP2318. This research is funded in part by the Gordon and Betty Moore Foundation's EPiQS Initiative, Grant GBMF8683 to T.S. M.R. acknowledges support by the Austrian Science Fund FWF within the DK-ALM (W1259-N27). J.K. acknowledges support from

the US Army Research Office (grant no. W911NF-21-1-0262).

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
