# Peer review of "Tensor Network Python (TeNPy) version 1"

_SciPost Physics Codebases, doi:SciPost Phys. Codebases 41 (2024) , SciPost Phys. Codebases 41-r1.0 (2024)_

## Round 1 · Referee Report · Anonymous (Referee 1) · 2024-9-10

Report

The TenPy package is well-known and heavily used in the community. This publication is a straightforward extension of the recent release of Version 1.

I don't see any weaknesses in this manuscript, so I recommend publication as it is.

Recommendation

Publish (meets expectations and criteria for this Journal)

---

## Round 1 · Referee Report · Anonymous (Referee 2) · 2024-10-4

Report

This is an excellent paper about the TeNPy software. TeNPy fills a number of needs in the community by offering a state-of-the-art tensor network library with a Python interface and offering features such as scriptable input files for reproducability, among other features.

The paper satisfies all of the requirements for SciPost Codebases, including code examples, benchmarks, and extensive documentation. The forum on the TeNPy website is another strong aspect of the library.

I verified that the software could be installed and run as specified in the paper, and I ran one of the examples in the paper to confirm that the results were correct.

Requested changes

A small clarification I wanted to ask about is on page 6, where it says that the code shown "discards singular values below $10^{−10}$". Are the singular values discarded based on a relative weight (such as dividing them by the largest singular value or a weight involving the sum of squares of singular values) or just based on their bare value? If the former, it would be good to be more precise here. If it is the bare value, then the current text is good.

A very minor correction is that in the legend of Figure 5, "iTensor" should be written as "ITensor" with a capital "I".

The main improvement needed to the paper would be to add a paragraph or so clarifying the relationship of TeNPY to NumPy and especially the tenpy.linalg layer to NumPy. Do I understand correctly that the dense tensors used by TeNPy are NumPy tensors, but then also that tenpy.linalg.np_conserved is a block-sparse tensor type that offers a compatible inferface to NumPy tensors? I think the paper would benefit from spelling out this connection in a bit more detail, to explain what experience users will have if they want to drop down to the layer of working directly with tensors, to implement an exotic type of entanglement measure, for example.

Recommendation

Ask for minor revision

  • validity: top
  • significance: top
  • originality: top
  • clarity: top
  • formatting: perfect
  • grammar: perfect

Author:  Johannes Hauschild  on 2024-10-28  [id 4906]

(in reply to Report 2 on 2024-10-04)
Category:
answer to question

We thank the referee for taking the time to prepare a report with helpful feedback. We will incorporate the suggested changes in the final manuscript.

The singular values in TeNPy are always truncated after normalization to $\sum_i
\Lambda_i ^2 = 1$. We nevertheless added a short comment clarifying this.

TeNPy indeed uses the NumPy library for the individual blocks of a tensor. It implements a higher-level tenpy.linalg.np_conserved.Array representing a Tensor with charge conservation. Only the functions which need to be optimized, e.g. because they loop over many small blocks inside the Array, are written in Cython (in tenpy/linalg/_npc_helper.pyx), which gets translated to C++ and then compiled during the installation process.

For version 2.0 of TeNPy, we are working on a pure C++ implementation, rewriting this whole part.

---

## Round 1 · Referee Report · Anonymous (Referee 3) · 2024-10-15

Report

The authors present a technical manuscript accompanying version 1.0 of the tenpy library, a widely used Python package to simulate matrix product states. The package strikes an excellent balance between allowing low-level access and a beginner friendly interface.

The tenpy library is available on github and can be readily installed via pip and conda.
I successfully tested the installation via pip and additionally successfully ran all tests provided in the repository. It is commendable that the authors follow well-established software development guidelines and additionally offer user support via a forum.

The tenpy package is a cornerstone of open-source tensor network simulation software. It compares well to other libraries like QUIMB (Python), ITensor (Julia) or TensorKit.jl (Julia).

The manuscript is well-written and easy to understand. It excellently suited to the criteria of SciPost Physics Codebase submissions as it fulfills all of the acceptance criteria. The authors take great care to explain the code that is presented in two examples (time-evolution in the Heisenberg case and ground state properties of the transverse field Ising model).

I recommend publication with (optional) revisions (see requested changes below).

Requested changes

  1. Introduction, Page 1: "where the infamous sign problem...": I suggest to include a citation here for the interested reader.
  2. Package Overview, Page 3: In Python, the words "package","module", "subpackage" and "submodule" are not interchangeable. A module usually refers to a single Python file, while a package is a collection of modules, usually together with an init.py file by (see, e.g. https://stackoverflow.com/questions/63906100/python-module-vs-sub-module-vs-package-vs-sub-package). Since this is a technical manuscript about a Python package, I recommend to either use the established terminology or to clarify that you are going to speak about it in more general, software-architecture terms.

Recommendation

Publish (easily meets expectations and criteria for this Journal; among top 50%)

  • validity: top
  • significance: high
  • originality: ok
  • clarity: high
  • formatting: perfect
  • grammar: perfect

Author:  Johannes Hauschild  on 2024-10-28  [id 4907]

(in reply to Report 3 on 2024-10-15)

We thank the referee for taking the time to prepare a report with helpful feedback. We will incorporate the suggested changes in the final manuscript.

  1. We added a citation to Troyer 2005.
  2. We went through the manuscript and adopt the notation to the one used on https://docs.python.org/3/tutorial/modules.html

---

## Editorial Decision

published